# ADA-BOUNDARY: ACCELERATING THE DNN TRAINING VIA ADAPTIVE BOUNDARY BATCH SELECTION

## ABSTRACT

Neural networks can converge faster with help from a smarter batch selection strategy. In this regard, we propose *Ada-Boundary*, a novel adaptive-batch selection algorithm that constructs an effective mini-batch according to the learning progress of the model. Our key idea is to present confusing samples what the true label is. Thus, the samples near the current decision boundary are considered as the most effective to expedite convergence. Taking advantage of our design, *Ada-Boundary* maintains its dominance in various degrees of training difficulty. We demonstrate the advantage of *Ada-Boundary* by extensive experiments using two convolutional neural networks for three benchmark data sets. The experiment results show that *Ada-Boundary* improves the training time by up to 31.7% compared with the state-of-the-art strategy and by up to 33.5% compared with the baseline strategy.

## 1 INTRODUCTION

Deep neural networks (DNNs) have achieved remarkable performance in many fields, especially, in computer vision and natural language processing (Krizhevsky et al., 2012; Goodfellow et al., 2016). Nevertheless, as the size of data grows very rapidly, the training step via stochastic gradient descent (SGD) based on mini-batches suffers from extremely high computational cost, which is mainly due to *slow convergence*. The common approaches for expediting convergence include some SGD variants (Zeiler, 2012; Kingma and Ba, 2015) that maintain individual learning rates for parameters and batch normalization (Ioffe and Szegedy, 2015) that stabilizes gradient variance.

Recently, in favor of the fact that not all samples have an equal impact on training, many studies have attempted to design sampling schemes based on the sample importance (Wu et al., 2017; Fan

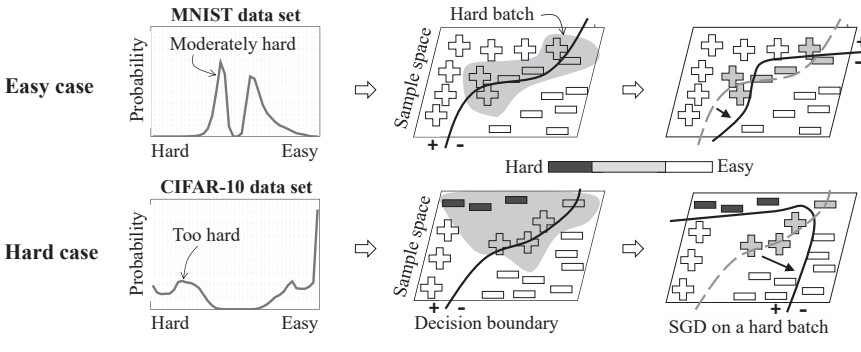

(a) Difficulty distribution.          (b) Hard sample oriented training.

Figure 1: Analysis on hard batch selection strategy: (a) shows the true sample distribution according to the difficulty computed by Eq. (1) at the training accuracy of 60%. An easy data set (MNIST) does not have "too hard" sample but "moderately hard" samples colored in gray, whereas a relatively hard data set (CIFAR-10) has many "too hard" samples colored in black. (b) shows the result of SGD on a hard batch. The moderately hard samples are informative to update a model, but the too hard samples make the model overfit to themselves.

et al., 2017; Katharopoulos and Fleuret, 2018). *Curriculum learning* (Bengio et al., 2009) inspired by human's learning is one of the representative methods to speed up the training step by gradually increasing the difficulty level of training samples. In contrast, deep learning studies focus on giving *higher* weights to *harder* samples during the entire training process. When the model requires a lot of epochs for convergence, it is known to converge faster with the batches of hard samples rather than randomly selected batches (Schaul et al., 2016; Loshchilov and Hutter, 2016; Gao and Jojic, 2017). There are various criteria for judging the hardness of a sample, e.g., the rank of the loss computed from previous epochs (Loshchilov and Hutter, 2016).

Here, a natural question arises: **Does the "hard" batch selection always speed up DNN training?** Our answer is **partially yes**: it is helpful only when training an *easy* data set. According to our in-depth analysis, as demonstrated in Figure 1(a), the *hardest* samples in a hard data set (e.g., CIFAR-10) were *too hard* to learn. They are highly likely to make the decision boundary bias towards themselves, as shown in Figure 1(b). On the other hand, in an easy data set (e.g., MNIST), the *hardest* samples, though they are just moderately hard, provide useful information for training. In practice, it was reported that hard batch selection succeeded to speed up only when training the easy MNIST data set (Loshchilov and Hutter, 2016; Gao and Jojic, 2017), and our experiments in Section 4.4 also confirmed the previous findings. This limitation calls for a new sampling scheme that supports both easy and hard data sets.

In this paper, we propose a novel adaptive batch selection strategy, called ***Ada-Boundary***, that accelerates training and is better generalized to hard data sets. As opposed to existing hard batch selection, *Ada-Boundary* picks up the samples with the most appropriate difficulty, considering the learning progress of the model. The samples near the *current* decision boundary are selected with high probability, as shown in Figure 2(a). Intuitively speaking, the samples far from the decision boundary are not that helpful since they are either too hard or too easy: those on the incorrect (or correct) side are too hard (or easy). This is the reason why we regard the samples around the decision boundary, which are *moderately hard*, as having the appropriate difficulty at the moment.

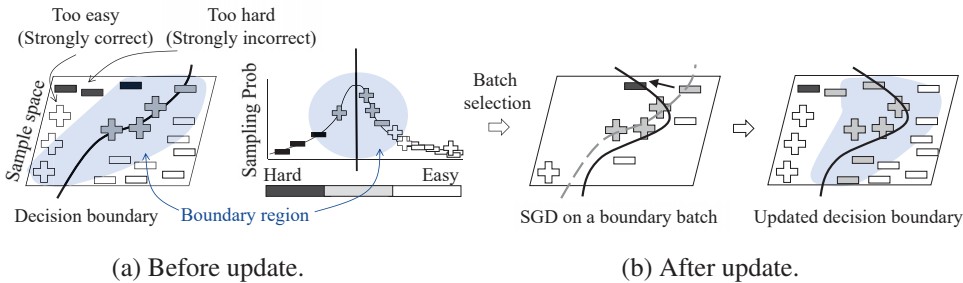

(a) Before update.  (b) After update.

Figure 2: Key idea of *Ada-Boundary*: (a) shows the sampling process of *Ada-Boundary*, (b) shows the results of an SGD iteration on the boundary samples.

Overall, the key idea of *Ada-Boundary* is to use the distance of a sample to the decision boundary for the hardness of the sample. The beauty of this design is *not* to require human intervention. The current decision boundary should be directly influenced by the learning progress of the model. The decision boundary of a DNN moves towards eliminating the incorrect samples as the training step progresses, so the difficulty of the samples near the decision boundary gradually increases as the model is learned. Then, the decision boundary keeps updated to identify the *confusing* samples in the middle of SGD, as illustrated in Figure 2(b). This approach is able to accelerate the convergence speed by providing the samples suited to the model at every SGD iteration, while it is less prone to incur an overfitting issue.

We have conducted extensive experiments to demonstrate the superiority of *Ada-Boundary*. Two popular convolutional neural network (CNN)[1] models are trained using three benchmark data sets. Compared to *random batch* selection, *Ada-Boundary* significantly reduces the execution time by 14.0–33.5%. At the same time, it provides a relative improvement of test error by 7.34–14.8% in the final epoch. Moreover, compared to the state-of-the-art hard batch selection (Loshchilov and Hutter,

---

[1]The idea is applicable to the DNNs other than CNNs, and we leave this extension as the future work.

2016), *Ada-Boundary* achieves the execution time smaller by $18.0\%$ and the test error smaller by $13.7\%$ in the CIFAR-10 data set.

## 2  *Ada-Boundary* COMPONENTS

The main challenge for *Ada-Boundary* is to evaluate how close a sample is to the decision boundary. In this section, we introduce a novel distance measure and present a method of computing the sampling probability based on the measure.

### 2.1  SAMPLE'S DISTANCE BASED ON SOFTMAX DISTRIBUTION

To evaluate the sample's distance to the decision boundary, we note that the softmax distribution, which is the output of the softmax layer in neural networks, clearly distinguishes how confidently the learner predicts and whether the prediction is right or wrong, as demonstrated in Figure 3.

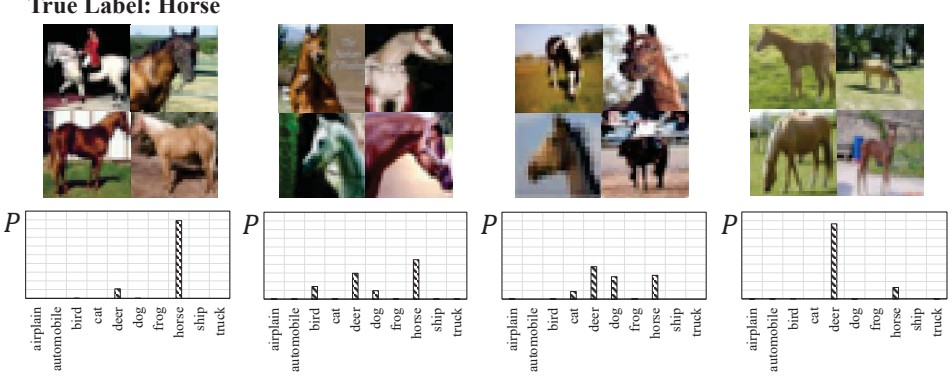

(a) Strongly correct.    (b) Weakly correct.    (c) Weakly incorrect.    (d) Strongly incorrect.

Figure 3: Classification of CIFAR-10 samples using the softmax distribution obtained from WideResNet 16-8 when training accuracy is $90\%$. If the prediction probability of the true label is the highest, the prediction is correct; otherwise, incorrect. If the highest probability dominates the distribution, the model's confidence is strong; otherwise, weak.

Let $h(y|x_i; \boldsymbol{\theta}^t)$ be the softmax distribution of a given sample $x_i$ over $y \in \{1, 2, \ldots, k\}$ labels, where $\boldsymbol{\theta}^t$ is the parameter of a neural network at time $t$. Then, the distance from a sample $x_i$ with the true label $y_i$ to the decision boundary of the neural network with $\boldsymbol{\theta}^t$ is defined by the directional distance function in Eq. (1). More specifically, the function consists of two terms related to the direction and magnitude of the distance, determined by the model's correctness and confidence, respectively. The correctness is determined by verifying whether the label with the highest probability matches the true label $y_i$, and the confidence is computed by the standard deviation of the softmax distribution. Intuitively, the standard deviation is a nice indicator of the confidence because the value gets closer to zero when the learner confuses.

$$dist(x_i, y_i; \boldsymbol{\theta}^t) = \overbrace{sign(x_i, y_i)}^{\text{correctness}} \cdot \overbrace{std(h(y|x_i; \boldsymbol{\theta}^t))}^{\text{confidence}}$$
$$sign(x_i, y_i) = \begin{cases} +1, & argmax_{y \in \{1,2,\ldots,k\}} h(y|x_i; \boldsymbol{\theta}^t) = y_i \\ -1, & otherwise \end{cases} \tag{1}$$

One might argue that the cross-entropy loss, $H(p, q) = -p(x_i) \log(q(x_i))$ where $p(x_i)$ and $q(x_i)$ are the true and softmax distributions for $x_i$, can be adopted for the distance function. However, because $p(x_i)$ is formulated as a one-hot true label vector, the cross-entropy loss cannot capture the prediction probability for *false* labels, which is an important factor of confusing samples.

Another advantage is that our distance function is *bounded* as opposed to the loss. For $k$ labels, the maximum value of $std(h(y|x_i; \boldsymbol{\theta}^t))$ is $k^{-1}\sqrt{(k-1)}$ when $h(m|x_i; \boldsymbol{\theta}^t) = 1$ and $\forall_{l \neq m} h(l|x_i; \boldsymbol{\theta}^t) = 0$. Thus, $dist(x_i, y_i; \boldsymbol{\theta}^t)$ is bounded as in Eq. (2).

$$-k^{-1}\sqrt{k-1} \leq dist(x_i, y_i; \boldsymbol{\theta}^t) \leq k^{-1}\sqrt{k-1} \tag{2}$$

## 2.2 Sampling Probability based on Quantization Index

The rank-based approach introduced by Loshchilov and Hutter (2016) is a common way to make the sampling probability of being selected for the next mini-batch. This approach sorts the samples by a certain importance measure in descending order, and exponentially decays the sampling probability of a given sample according to its rank. Let $N$ denote the total number of samples. Then, each $r$-th ranked sample is selected with the probability $p(r)$ which drops by a factor of $\exp\left(\log(s_e)/N\right)$. Here, $s_e$ is the *selection pressure* parameter that affects the probability gap between the most and the least important samples. When normalized to sum up to $1.0$, the probability of the $r$-th ranked sample's being selected is defined by Eq. (3).

$$p(r) = \frac{1/\exp\left(\log(s_e)/N\right)^r}{\sum_{j=1}^{N} 1/\exp\left(\log(s_e)/N\right)^j} \tag{3}$$

In the existing rank-based approach, the rank of a sample is determined by $|dist(x_i, y_i; \boldsymbol{\theta}^t)|$ in ascending order, because it is inversely proportional to the sample importance. However, if the mass of the true sample distribution is skewed to one side (e.g., easy side) as shown in Figure 4, the mini-batch samples are selected with high probability from the skewed side rather than around the decision boundary where $|dist(x_i, y_i; \boldsymbol{\theta}^t)|$ is very small. This problem was attributed to unconditionally fixed probability to a given rank. In other words, the samples with similar ranks are selected with similar probabilities regardless of the magnitude of the distance values.

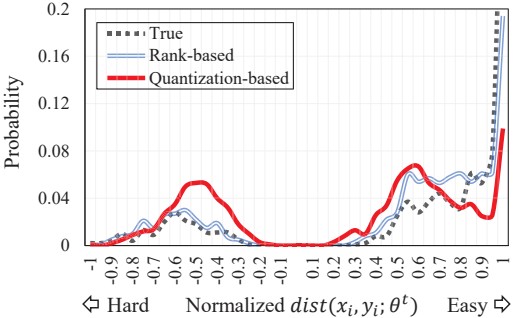

Figure 4: Sample distribution according to the normalized $dist(x_i, y_i; \boldsymbol{\theta}^t)$ at the training accuracy of $80\%$, when training LeNet-5 ($s_e = 100$) with the Fashion-MNIST data set. The distributions of mini-batch samples selected by the rank-based and quantization-based approaches, respectively, are plotted together with the true sample distribution.

To incorporate the impact of the distance into batch selection, we adopt the quantization method (Gray and Neuhoff, 1998; Chen and Wornell, 2001) and use the quantization index $q$ instead of the rank $r$. Let $\Delta$ be the quantization step size and $d$ be the output of the function $dist(x_i, y_i; \boldsymbol{\theta}^t)$ of a given sample $x_i$. Then, the index $q$ is obtained by the quantizer $Q(d)$ as in Eq. (4). The quantization index gets larger as a sample moves away from the decision boundary. In addition, the difference between two indexes reflects the difference in the actual distances.

$$q = Q(d), \quad Q(d) = \lceil |d|/\Delta \rceil \tag{4}$$

In Eq. (4), we set $\Delta$ to be $k^{-1}\sqrt{k-1}/N$ such that the index $q$ is bounded to $N$ (the total number of samples) by Eq. (2). The sampling probability of a given sample $x_i$ with the true label $y_i$ is defined as Eq. (5). As shown in Figure 4, our quantization-based method provides a *well-balanced* distribution, even if the true sample distribution is skewed.

$$p(x_i, y_i) = \frac{1/\exp\left(\log(s_e)/N\right)^{Q(dist(x_i, y_i; \boldsymbol{\theta}^t))}}{\sum_{j=1}^{N} 1/\exp\left(\log(s_e)/N\right)^{Q(dist(x_j, y_j; \boldsymbol{\theta}^t))}} \tag{5}$$

## 3 Ada-Boundary ALGORITHM

### 3.1 MAIN PROPOSED ALGORITHM

Algorithm 1 describes the overall procedure of Ada-Boundary. The input to the algorithm consists of the samples of size $N$ (i.e., training data set), the mini-batch size $b$, the selection pressure $s_e$, and the threshold $\gamma$ used to decide the warm-up period. In the early stages of training, since the quantization index for each sample is not confirmed yet, the algorithm requires the warm-up period during $\gamma$ epochs. Randomly selected mini-batch samples are used to warm-up (Lines 6–7), and their quantization indexes are updated (Lines 11–16). After the warm-up epochs, the algorithm computes the sampling probability of each sample by Eq. (5) and selects mini-batch samples based on the probability (Lines 8–10). Then, the quantization indexes are updated in the same way (Lines 11–16). Here, we compute the indexes using the model with $\theta^{t+1}$ after every SGD step rather than every epoch, in order to reflect the latest state of the model; besides, we asynchronously update the indexes of the samples only included in the mini-batch, to avoid the forward propagation of the entire samples which induces a high computational cost.

---

**Algorithm 1** *Ada-Boundary* Algorithm

INPUT: $N$ samples, $numEpoch$, $b$: mini-batch size, $s_e$: selection pressure, $\gamma$: warm-up period
1: $t \leftarrow 1$;
2: $\theta^t \leftarrow$ Initialize the model parameter;
3: $q\_dict \leftarrow \{\}$; /* Dictionary for quantization indexes */
4: **for** $i = 1$ **to** $numEpoch$ **do**
5:     **for** $j = 1$ **to** $N/b$ **do**
6:         **if** $i \leq \gamma$ **then** /* Warm-up */
7:             $\{(x_1, y_1), \dots, (x_b, y_b)\} \leftarrow$ Randomly select next mini-batch samples;
8:         **else** /* Adaptive batch selection */
9:             $prob\_table \leftarrow Compute\_Probability(q\_dict, s_e)$; /* By Eq. (5) */
10:            $\{(x_1, y_1), \dots, (x_b, y_b)\} \leftarrow$ Select next mini-batch samples based on $prob\_table$;
11:         $loss \leftarrow Get\_Loss(\{(x_1, y_1), \dots, (x_b, y_b)\}, \theta^t)$; /* Forward 1 */
12:         $\theta^{t+1} \leftarrow SGD\_Step(loss, \theta^t)$; /* Backward */
13:         /* Asynchronous update */
14:         $\{h(y|x_1; \theta^{t+1}), \dots, h(y|x_b; \theta^{t+1})\} \leftarrow Get\_Softmax(\{x_1, \dots, x_b\}, \theta^{t+1})$; /* Forward 2 */
15:         **for** $m = 1$ **to** $b$ **do**
16:            $q\_dict[x_m] = Q(dist(x_m, y_m; \theta^{t+1}))$; /* Compute quantization indexes by Eq. (4) */
17:     $t \leftarrow t + 1$;

---

### 3.2 VARIANTS OF *Ada-Boundary* FOR COMPARISON

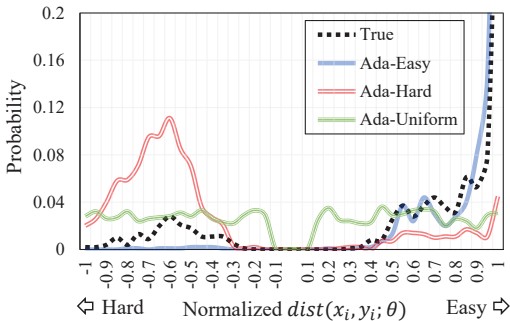

Figure 5: The distributions of mini-batch samples selected by the three variants in the same configuration as Figure 4.

For a more sophisticated analysis of sampling strategies, we modify a few lines of Algorithm 1 to present three heuristic sampling strategies, which are detailed in Appendix A. (i) *Ada-Easy* is designed to show the effect of easy samples on training, so it focuses on the samples far from the

decision boundary to the *positive* direction. (ii) *Ada-Hard* is similar to the existing hard batch strategy (Loshchilov and Hutter, 2016), but it uses our distance function instead of the loss. That is, *Ada-Hard* focuses on the samples far from the decision boundary to the *negative* direction, which is the opposite of *Ada-Easy*. (iii) *Ada-Uniform* is designed to select the samples for a wide range of difficulty, so it samples uniformly over the distance range regardless of the sample distribution. Figure 5 shows the distributions of mini-batch samples drawn by these three variants. The distribution of *Ada-Easy* is skewed to the easy side, that of *Ada-Hard* is skewed to the hard side, and that of *Ada-Uniform* tends to be uniform.

To avoid additional inference steps of *Ada-Boundary* (Line 14 in Algorithm 1), we present a history-based variant, called *Ada-Boundary(History)*. It updates the qunatization indexes using the *previous* model with $\theta^t$. See Appendix B for the detailed algorithm and experiment results.

## 4 EVALUATION

### 4.1 DATA SETS AND ARCHITECTURES

In this section, all the experiments were performed on three benchmark data sets: MNIST[2] of handwritten digits (LeCun, 1998) with 60,000 training and 10,000 testing images; Fashion-MNIST[3] of various clothing (Xiao et al., 2017) with 60,000 training and 10,000 testing images; and CIFAR-10[4] of a subset of 80 million categorical images (Krizhevsky et al., 2014) with 50,000 training and 10,000 testing images. We did not apply any data augmentation and pre-processing procedures.

A simple model LeNet-5 (LeCun et al., 2015) was used for two easy data sets, MNIST and Fasion-MNIST. A complex model WideResNet-16-8 (Zagoruyko and Komodakis, 2016) was used for a relatively difficult data set, CIFAR-10. Batch normalization (Ioffe and Szegedy, 2015) was applied to both models. As for hyper-parameters, we used a learning rate of 0.01 and a batch size of 128; the training epoch was set to be 50 for LeNet-5 and 70 for WideResNet-16-8, which is early stopping to clearly show the difference in convergence speed. Regarding those specific to our algorithm, we set the selection pressure $s_e$ to be 100, which is the best value found from $s_e = \{10, 100, 1000\}$ on the three data sets, and set the warm-up threshold $\gamma$ to be 10. Technically, a small $\gamma$ was enough to warm-up, but to reduce the performance variance caused by randomly initialized parameters, we used the larger $\gamma$ and shared model parameters for all strategies during the warm-up period.

Due to the lack of space, the experimental results using DenseNet ($L = 25$, $k = 12$) (Huang et al., 2017) on two hard data sets, CIFAR-100[4] and Tiny-ImageNet[5], are discussed in Appendix C together with the impact of the selection pressure $s_e$.

### 4.2 ALGORITHMS

We compared *Ada-Boundary* with not only *random batch* selection but also four different adaptive batch selections. *Random batch* selection selects the next batch uniformly at random from the entire data set. One of four adaptive selections is the state-of-the-art strategy that selects hard samples based on the loss-rank, which is called *online batch* selection (Loshchilov and Hutter, 2016), and the remainders, *Ada-Easy*, *Ada-Hard*, and *Ada-Uniform*, are the three variants introduced in Section 3.2. All the algorithms were implemented using TensorFlow[6] and executed using a single NVIDIA Tesla V100 GPU on DGX-1. For reproducibility, we provide the source code at `https://github.com/anonymized`.

### 4.3 EVALUATION METRICS

To measure the performance gain over the baseline (*random batch* selection) as well as the state-of-art (*online batch* selection), we used the following three metrics. We repeated every test *five* times for robustness and reported the average. The wall-clock training time is discussed in Appendix D.

---

[2]`http://yann.lecun.com/exdb/mnist`
[3]`https://github.com/zalandoresearch/fashion-mnist`
[4]`https://www.cs.toronto.edu/~kriz/cifar.html`
[5]`https://tiny-imagenet.herokuapp.com/`
[6]`https://www.tensorflow.org/versions/r1.8/`

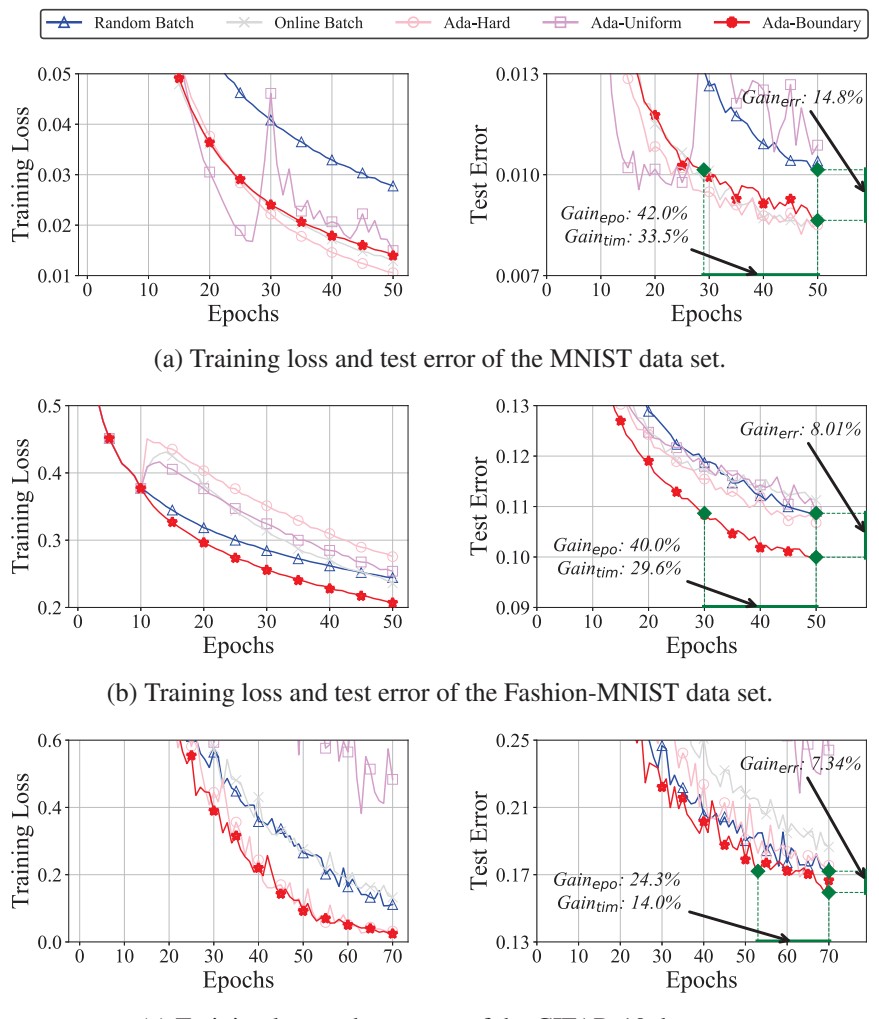

(a) Training loss and test error of the MNIST data set.

(b) Training loss and test error of the Fashion-MNIST data set.

(c) Training loss and test error of the CIFAR-10 data set.

Figure 6: Convergence curves of five batch selection strategies with SGD on three data sets.

(i) $Gain_{err}$: Reduction in *test error* at the final epoch (%). In Figure 6(a), at the 50th epoch, the test error of *random batch* selection was $1.014 \cdot 10^{-2}$, and that of *Ada-Boundary* was $8.643 \cdot 10^{-3}$. Thus, $Gain_{err}$ was $(1.014 \cdot 10^{-2} - 8.643 \cdot 10^{-3})/1.014 \cdot 10^{-2} \times 100 = 14.8\%$.

(ii) $Gain_{epo}$: Reduction in number of *epochs* to obtain the same error (%). In Figure 6(a), the test error of $1.014 \cdot 10^{2}$ achieved at the 50th epoch by *random batch* selection can be achieved only at the 29th epoch by *Ada-Boundary*. Thus, $Gain_{err}$ was $(50 - 29)/50 \times 100 = 42.0\%$.

(iii) $Gain_{tim}$: Reduction in *running time* to obtain the same error (%). In Figure 6(a), similar to $Gain_{epo}$, $Gain_{tim}$ was $(205.0 - 136.3)/205.0 \times 100 = 33.5\%$.

## 4.4 CONVERGENCE ANALYSIS

Figure 6 shows the convergence curves of training loss and test error for five batch selection strategies on three data sets, when we used the SGD optimizer for training. In order to improve legibility, only the curves for the baseline and proposed strategies are dark colored; thus, the three metrics in the figure were calculated against the baseline strategy, *random batch* selection. Owing to the lack of space, we discuss the results with the momentum optimizer in Appendix E. *Ada-Easy* was excluded in Figure 6 because its convergence speed was much slower than other strategies. That is, easy samples did not contribute to expedite training. We conduct convergence analysis of the five batch selection strategies for the *same number of epochs*, as follows:

- **MNIST** (Figure 6(a)): All adaptive batch selections achieved faster convergence speed compared with *random batch* selection. *Ada-Boundary*, *Ada-Hard*, and *online batch* selection showed similar performance. *Ada-Uniform* was the fastest at the beginning, but its training loss and test error increased sharply in the middle of the training or testing procedures.

- **Fashion-MNIST** (Figure 6(b)): *Ada-Boundary* showed the fastest convergence speed in both training loss and test error. In contrast, after warm-up epochs, the training loss of the other adaptive batch selections increased temporarily, and their test error at the final epoch became similar to that of *random batch* selection.

- **CIFAR-10** (Figure 6(c)): *Ada-Boundary* and *Ada-Hard* showed the fastest convergence on training loss, but in test error, the convergence speed of *Ada-Hard* was much slower than that of *Ada-Boundary*. This means that focusing on hard samples results in the overfitting to "too hard" samples, which is indicated by a larger difference between the converged training loss (error) and the converged test error. Also, the slow convergence speed of *online batch* selection in test error is explained by the same reason.

In summary, in the easiest MNIST data set, all adaptive batch selections accelerated their convergence speed compared with *random batch* selection. However, as the training difficulty (complexity) increased from MNIST to Fashion-MNIST and further to CIFAR-10, only *Ada-Boundary* converged significantly (by $Gain_{err}$) faster than *random batch* selection.

## 4.5 SUMMARY OF PERFORMANCE GAINS

We clarify the quantitative performance gains of *Ada-Boundary* over *random batch* and *online batch* selections in Table 1. *Ada-Boundary* significantly outperforms both strategies, as already shown in Figure 6. There is only one exception in MNIST, because *online batch* selection is known to work well with an easy data set (Loshchilov and Hutter, 2016). The noticeable advantage of *Ada-Boundary* is to reduce the training time significantly by up to around $30\%$, which is really important for huge, complex data sets.

Table 1: Performance gains over *random batch* and *online batch* selections.

| Comparison target | Against random batch selection | | | Against online batch selection | | |
|---|---|---|---|---|---|---|
| Metrics | $Gain_{err}$ | $Gain_{epo}$ | $Gain_{tim}$ | $Gain_{err}$ | $Gain_{epo}$ | $Gain_{tim}$ |
| MNIST | 14.8% | 42.0% | 33.5% | −2.08% | 0.00% | 0.00% |
| Fashion-MNIST | 8.01% | 40.0% | 29.6% | 10.2% | 42.0% | 31.7% |
| CIFAR-10 | 7.34% | 24.3% | 14.0% | 13.7% | 46.0% | 18.0% |

## 5 RELATED WORK

There have been numerous attempts to understand which samples contribute the most during training. Curriculum learning (Bengio et al., 2009), inspired by the perceived way that humans and animals learn, first takes easy samples and then gradually increases the difficulty of samples in a manual manner. Self-paced learning (Kumar et al., 2010) uses the prediction error to determine the easiness of samples in order to alleviate the limitation of curriculum learning. They regard that the importance is determined by how easy the samples are. However, easiness is not sufficient to decide when a sample should be introduced to a learner (Gao and Jojic, 2017).

Recently, Tsvetkov et al. (2016) used Bayesian optimization to optimize a curriculum for training dense, distributed word representations. Sachan and Xing (2016) emphasized that the right curriculum not only has to arrange data samples in the order of difficulty, but also introduces a small number of samples that are dissimilar to the previously seen samples. Shrivastava et al. (2016) proposed a hard-example mining algorithm to eliminate several heuristics and hyper-parameters commonly used to select hard examples. However, these algorithms are designed to support only a designated task, such as natural language processing or region-based object detection. The neural data filter proposed by Fan et al. (2017) is orthogonal to our work because it aims at filtering the redundant samples from streaming data. As mentioned earlier, *Ada-Boundary* in general follows the philosophy of curriculum learning.

More closely related to the adaptive batch selection, Loshchilov and Hutter (2016) keep the history of losses for previously seen samples, and compute the sampling probability based on the loss rank. The sample probability to be selected for the next mini-batch is exponentially decayed with its rank. This allows the samples with low ranks (i.e., high losses) are considered more frequently for the next mini-batch. Gao and Jojic (2017)'s work is similar to Loshchilov and Hutter (2016)'s work except that gradient norms are used instead of losses to compute the probability. In contrast to curriculum learning, both methods focus on only hard samples for training. Also, they ignore the difference in actual losses or gradient norms by transforming the values to ranks. We have empirically verified that *Ada-Boundary* outperforms *online batch* selection (Loshchilov and Hutter, 2016), which is regarded as the state-of-the-art of this category. Similar to our work, Chang et al. (2017) claimed that the uncertain samples should be preferred during training, but their main contribution lies on training more accurate and robust model by choosing samples with high prediction variances. In contrast, our main contribution lies on training faster using confusing samples near the decision boundary.

For the completeness of the survey, we mention the work to accelerate the optimization process of conventional algorithms based on importance sampling. Needell et al. (2014) re-weight the obtained gradients by the inverses of their sampling probabilities to reduce the variance. Schmidt et al. (2015) biased the sampling to the Lipschitz constant to quickly find the solution of a strongly-convex optimization problem arising from the training of conditional random fields.

## 6 CONCLUSION AND FUTURE WORK

In this paper, we proposed a novel adaptive batch selection algorithm, *Ada-Boundary*, that presents the most appropriate samples according to the learning progress of the model. Toward this goal, we defined the distance from a sample to the decision boundary and introduced a quantization method for selecting the samples near the boundary with high probability. We performed extensive experiments using two CNN models for three benchmark data sets. The results showed that *Ada-Boundary* significantly accelerated the training process as well as was better generalized in hard data sets. When training an easy data set, *Ada-Boundary* showed a fast convergence comparable to that of the state-of-the-art algorithm; when training relatively hard data sets, only *Ada-Boundary* converged significantly faster than *random batch* selection.

The most exciting benefit of *Ada-Boundary* is to save the time needed for the training of a DNN. It becomes more important as the size and complexity of data becomes higher, and can be boosted with recent advance of hardware technologies. Our immediate future work is to apply *Ada-Boundary* to other types of DNNs such as the recurrent neural networks (RNN) (Mikolov et al., 2010) and the long short-term memory (LSTM) (Hochreiter and Schmidhuber, 1997), which have a neural structure completely different from the CNN. In addition, we plan to investigate the relationship between the power of a DNN and the improvement of *Ada-Boundary*.

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

## A    IMPLEMENTATION OF THE THREE VARIANTS

For *Ada-Easy* which prefers easy samples to hard samples, $q$ should be small for the sample located deep in the positive direction. For *Ada-Hard*, $q$ should be small for the sample located deep in the negative direction. Thus, *Ada-Easy* and *Ada-Hard* can be implemented by modifying the quantizers $Q(d)$ in Line 16 of Algorithm 1. When we set $\Delta = k^{-1}\sqrt{k-1}/N$ to make the index $q$ bound to $N$, the quantizers of *Ada-Easy* and *Ada-Hard* are defined as Eqs. (6) and (7), respectively.

$$q = Q(d)$$
$$Q(d) = \begin{cases} -\lceil d/2\Delta \rceil + N/2 + 1, & if \ d \geq 0 \\ -\lfloor d/2\Delta \rfloor + N/2, & otherwise \end{cases} \tag{6}$$

$$q = Q(d)$$
$$Q(d) = \begin{cases} \lceil d/2\Delta \rceil + N/2, & if \ d \geq 0 \\ \lfloor d/2\Delta \rfloor + N/2 + 1, & otherwise \end{cases} \tag{7}$$

*Ada-Uniform* can be implemented by using $F^{-1}(x)$ to compute the sampling probability in Line 9 of Algorithm 1, where $F(x)$ is the empirical sample distribution according to the sample's distance to the decision boundary.

## B    HISTORY-BASED *Ada-Boundary* VARIANT

We present *Ada-Boundary(History)* that updates the quantization indexes based on the *previous* model with $\theta^t$ instead of the latest model with $\theta^{t+1}$. This is easily accomplished by replacing Lines 11–17 of Algorithm 1 with those of Algorithm 2. *Ada-Boundary(History)* reduces the time required for additional inference steps that reflect the latest state of the model, which correspond to Lines 13–16 of Algorithm 1, at the expense of slight increase of test error.

---

**Algorithm 2** *Ada-Boundary(History)*

---

INPUT: $N$ samples, $numEpoch$, $b$: mini-batch size, $s_e$: selection pressure, $\gamma$: warm-up period
1: $t \leftarrow 1$;
2: $\theta^t \leftarrow$ Initialize the model parameter;
3: $q\_dict \leftarrow \{\}$; /* Dictionary for quantization indexes */
4: **for** $i = 1$ **to** $numEpoch$ **do**
5:     **for** $j = 1$ **to** $N/b$ **do**
6:         **if** $i \leq \gamma$ **then** /* Warm-up */
7:             $\{(x_1, y_1), \ldots, (x_b, y_b)\} \leftarrow$ Randomly select next mini-batch samples;
8:         **else** /* Adaptive batch selection */
9:             $prob\_table \leftarrow Compute\_Probability(q\_dict, s_e)$; /* By Eq. (5) */
10:             $\{(x_1, y_1), \ldots, (x_b, y_b)\} \leftarrow$ Select next mini-batch samples based on $prob\_table$;
11:         /* Forward and asynchronous update */
12:         $\{h(y|x_1; \theta^t), ..., \{h(y|x_b; \theta^t)\}, loss \leftarrow Get\_Softmax \& Loss(\{(x_1, y_1), ..., (x_b, y_b)\}, \theta^t)$;
13:         **for** $m = 1$ **to** $b$ **do**
14:             $q\_dict[x_m] = Q(dist(x_m, y_m; \theta^t))$; /* Compute quantization indexes by Eq. (4) */
15:         /* Backward */
16:         $\theta^{t+1} \leftarrow SGD\_Step(loss, \theta^t)$;
17:         $t \leftarrow t + 1$;

---

Figure 7 and Figure 8 show the convergence curves of training loss and test error for *Ada-Boundary(History)*, when we used the SGD and momentum optimizers, respectively.

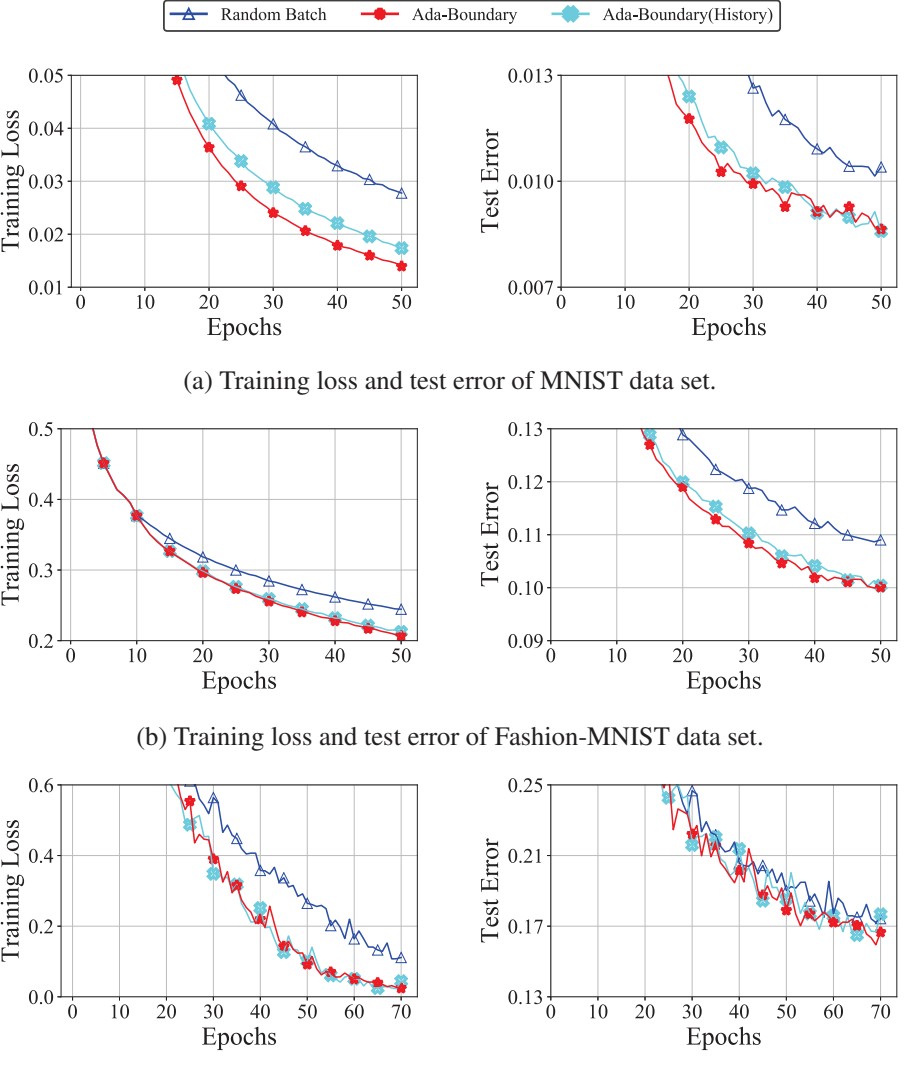

(a) Training loss and test error of MNIST data set.

(b) Training loss and test error of Fashion-MNIST data set.

(c) Training loss and test error of CIFAR-10 data set.

Figure 7: Convergence curves of *Ada-Boundary(History)* with **SGD** on three data sets.

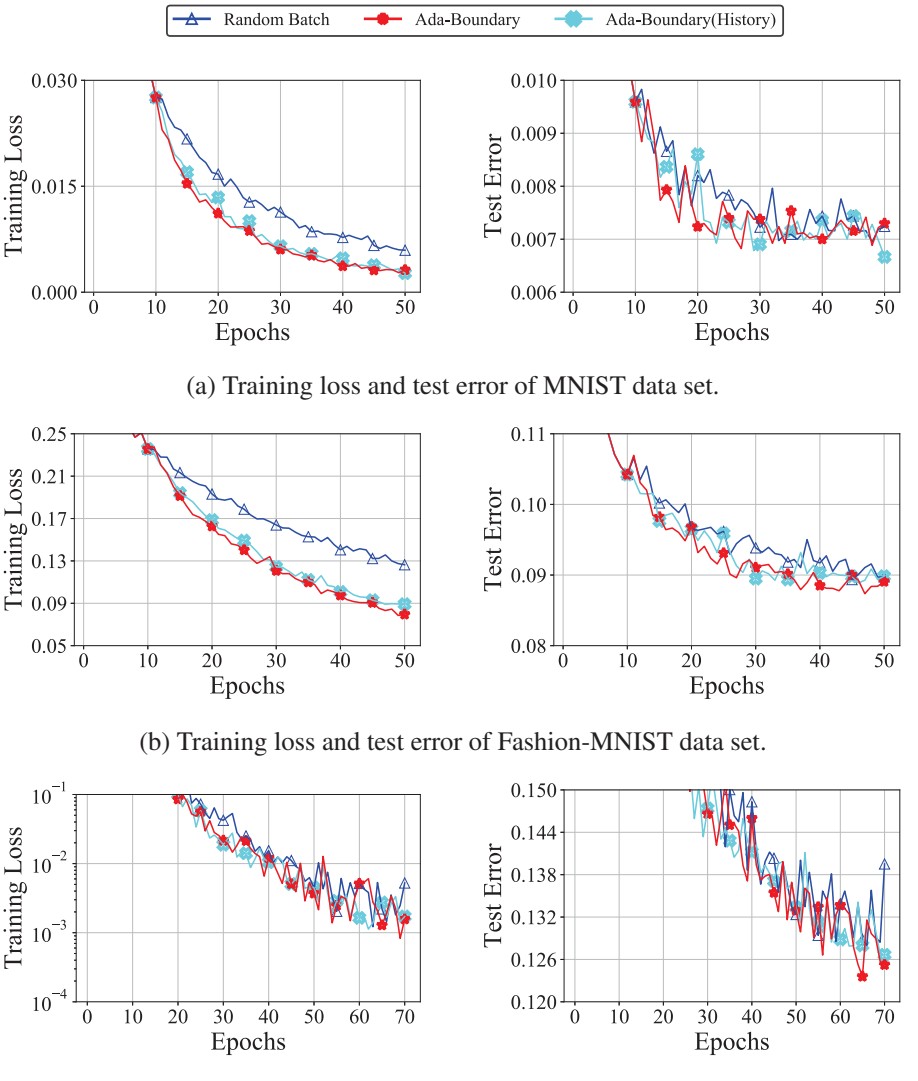

(a) Training loss and test error of MNIST data set.

(b) Training loss and test error of Fashion-MNIST data set.

(c) Training loss and test error of CIFAR-10 data set.

Figure 8: Convergence curves of *Ada-Boundary(History)* with **momentum** on three data sets.

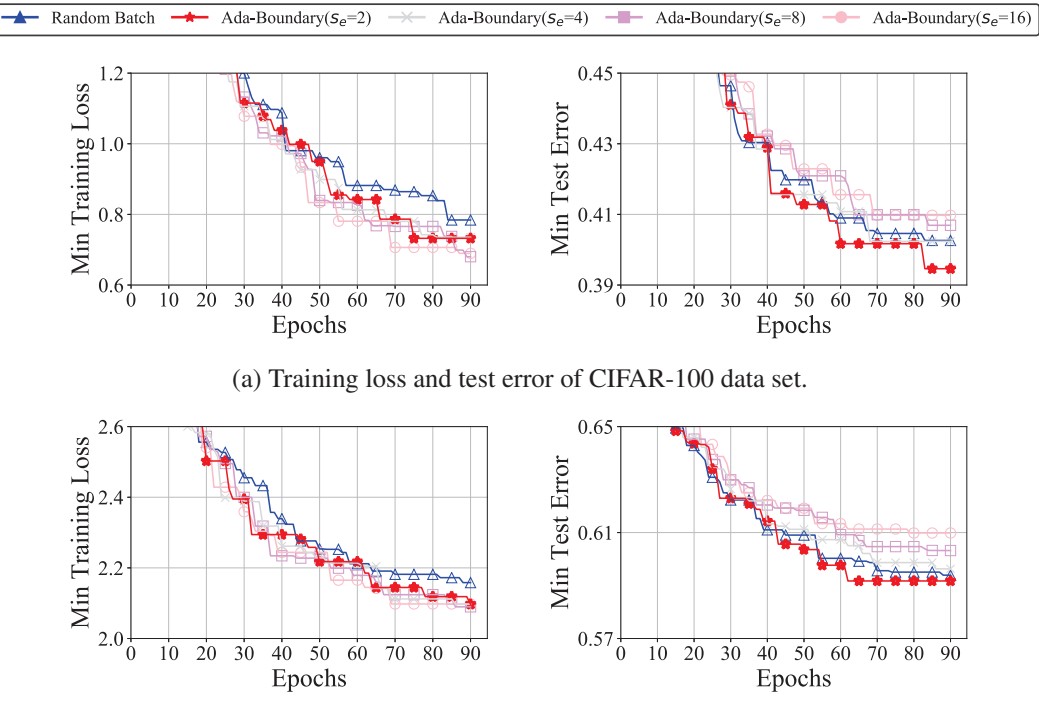

(a) Training loss and test error of CIFAR-100 data set.

(b) Training loss and test error of Tiny-ImageNet data set.

Figure 9: Convergence curves of *Ada-Boundary* with varying $s_e$ on two hard data sets.

## C  *Ada-Boundary* ON TWO HARD DATA SETS

As a practical paper, we include the experimental results on two more challenging data sets: CIFAR-100 composed of 100 image classes with $50,000$ training and $10,000$ testing images; Tiny-ImageNet composed of 200 image classes with $100,000$ training and $10,000$ testing images. All images in Tiny-ImageNet were resized to $32 \times 32$ images.

One of the state-of-the-art model DenseNet ($L$=25, $k$=12) (Huang et al., 2017) was used for two hard data sets with momentum optimizer. Regarding algorithm parameters, we used a learning rate of 0.1 and a batch size of 128; The training epoch and warm-up threshold $\gamma$ were set to be 90 and 10, respectively. We repeated every test *five* times for robustness and reported the average.

### C.1  IMPACT OF SELECTION PRESSURE $s_e$

The selection pressure $s_e$ determines how strongly the boundary samples are selected. The greater the $s_e$, the greater the sampling probability of the boundary sample, so more boundary samples were chosen for the next mini-batch. On the other hand, the less $s_e$ makes *Ada-Boundary* closer to *random batch* selection.

Figure 9 shows the convergence curves of *Ada-Boundary* with varying $s_e$ on two hard data sets. To clearly analyze the impact of the selection pressure, we plotted the minimum of training loss and test error with a given epochs. Overall, the convergence speed of training loss was accelerated as the $s_e$ increased from 2 to 16, but that of test error was faster only when the $s_e$ was less than a certain value. The convergence speed of test error was faster than *random batch* selection, when $s_e$ was less than or equal to 4 (CIFAR-100) and 2 (Tiny-ImageNet). Surprisingly, the overexposure to the boundary samples using the large $s_e$ incurred the overfitting issue in hard data sets, whereas the large $s_e = 100$ worked well for our easy or relatively hard data sets as discussed in Section 4. That is, the selection pressure $s_e$ should be chosen more carefully considering the difficulty of the given data set. We leave this challenge as our future work.

## C.2 Performance Analysis

Table 2 shows the performance gains of *Ada-Boundary* over *random batch* selection on two hard data sets. We only quantify the gains of *Ada-Boundary*($s_e = 2$) because its performance was the best as shown in Figure 9. *Ada-Boundary*($s_e = 2$) always outperforms *random batch* selection. Especially, it reduces the training time significantly by up to around 20%.

Table 2: Performance gains of *Ada-Boundary*($s_e = 2$) over random batch selection in Figure 9.

| Comparison target | Against random batch selection | | |
|---|---|---|---|
| Metrics | $Gain_{err}$ | $Gain_{epo}$ | $Gain_{tim}$ |
| CIFAR-100 | 1.99% | 33.3% | 21.4% |
| TINY-ImageNet | 0.37% | 31.1% | 18.0% |

Table 3 shows the wall-clock training time for the same number of parameter updates on two hard data sets (Figure 9). *Ada-Boundary*($s_e = 2$) with momentum was 15.2%–16.0% slower than *random batch* selection. However, it reduced the running time by 18.0%–21.4% (by $Gain_{tim}$) to obtain the same test error of *random batch* selection.

Table 3: Wall-clock training time for Figure 9 (seconds).

| Optimizer | Momentum (Figure 9) | |
|---|---|---|
| Data sets | CIFAR-100 | Tiny-ImageNet |
| Random batch | 1917 | 3814 |
| *Ada-Boundary*($s_e = 2$) | 2260 | 4542 |

# D  Wall-clock Training Time

The procedures for recomputing sampling probabilities and updating quantization indexes make *Ada-Boundary* slower than *random batch* selection. Table 4 shows the wall-clock training time for the same number of parameter updates (i.e., the same number of epochs) with SGD (Figure 6) and momentum (Figure 10). *Ada-Boundary* with SGD was 12.8%–14.7% and 6.06%–12.2% slower than *random batch* and *online batch* selections, respectively. *Ada-Boundary* with momentum was 13.1%–14.7% and 6.67%–12.2% slower than *random batch* and *online batch* selections, respectively. Although *Ada-Boundary* took longer for the same number of updates, *Ada-Boundary* achieved significant reduction in running time by 7.96%–33.5% (by $Gain_{tim}$) to obtain the same test error of *random batch* selection due to the fast convergence.

Table 4: Wall-clock training time for Figure 6 and Figure 10 (seconds).

| Optimizer | SGD (Figure 6) | | | Momentum (Figure 10) | | |
|---|---|---|---|---|---|---|
| Data sets | MNIST | Fashion-MNIST | CIFAR-10 | MNIST | Fashion-MNIST | CIFAR-10 |
| Random batch | 205 | 197 | 3347 | 199 | 192 | 3355 |
| Online batch | 218 | 217 | 3371 | 211 | 210 | 3388 |
| Ada-Boundary | 235 | 231 | 3838 | 231 | 225 | 3860 |

# E  Experiment Results using Momentum Optimizer

## E.1 Convergence Analysis

Figure 10 shows the convergence curves of training loss and test error for five batch selection strategies on three data sets, when we used the momentum optimizer with setting the momentum to be 0.9. In the MNIST data set, we limited the number of epochs to be 30 because both training loss and test error were fully converged after 30 epochs. We repeat the convergence analysis, as follows:

- **MNIST** (Figure 10(a)): Except *Ada-Uniform*, all adaptive batch selections converged faster than *random batch* selection. *Online batch* selection showed much faster convergence speed than other adaptive batch selections in training loss, but converged similarly with the others in test error owing to the overfitting to hard samples.

- **Fashion-MNIST** (Figure 10(b)): *Ada-Boundary* showed the fastest convergence speed in test error, although it did not converge faster than *online batch* selection in training loss. In contrast, *online batch* selection was the fastest in training loss, but its convergence in test error was slightly slower than that of *random batch* selection. This emphasizes the need to consider the samples with appropriate difficulty rather than hard samples. The convergence speeds of *Ada-Hard* and *Ada-Uniform* in test error were slower than that of *random batch* selection.

- **CIFAR-10** (Figure 10(c)): In both training loss and test error, *Ada-Boundary* and *Ada-Hard* showed slightly faster convergence speed than *random batch* selection. On the other hand, *online batch* selection converged slightly slower than *random batch* selection in both cases.

In summary, in the easiest MNIST data set, most of adaptive batch selections accelerated their convergence speed compared with *random batch* selection. However, in Fashion-MNIST data set, only *Ada-Boundary* converged faster than *random batch* selection. In a relatively difficult CIFAR-10 data set, *Ada-Boundary* and *Ada-Hard* showed comparable convergence speed and then converged faster than *random batch* selection.

### E.2 SUMMARY OF PERFORMANCE GAINS

We quantify the performance gains of *Ada-Boundary* over *random batch* and *online batch* selections in Table 5. *Ada-Boundary* always outperforms both strategies, as already shown in Figure 10. Compared with Table 1, $Gain_{tim}$ over *random batch* selection tends to become smaller, whereas $Gain_{tim}$ over *online batch* selection tends to become larger.

Table 5: Performance gains over two existing strategies in Figure 10.

| Comparison target | Against random batch selection | | | Against online batch selection | | |
|---|---|---|---|---|---|---|
| Metrics | $Gain_{err}$ | $Gain_{epo}$ | $Gain_{tim}$ | $Gain_{err}$ | $Gain_{epo}$ | $Gain_{tim}$ |
| MNIST | 5.58% | 26.7% | 14.9% | 2.27% | 13.0% | 4.75% |
| Fashion-MNIST | 2.24% | 28.0% | 15.6% | 4.54% | 46.0% | 42.1% |
| CIFAR-10 | 3.43% | 20.0% | 7.96% | 4.02% | 28.0% | 18.0% |

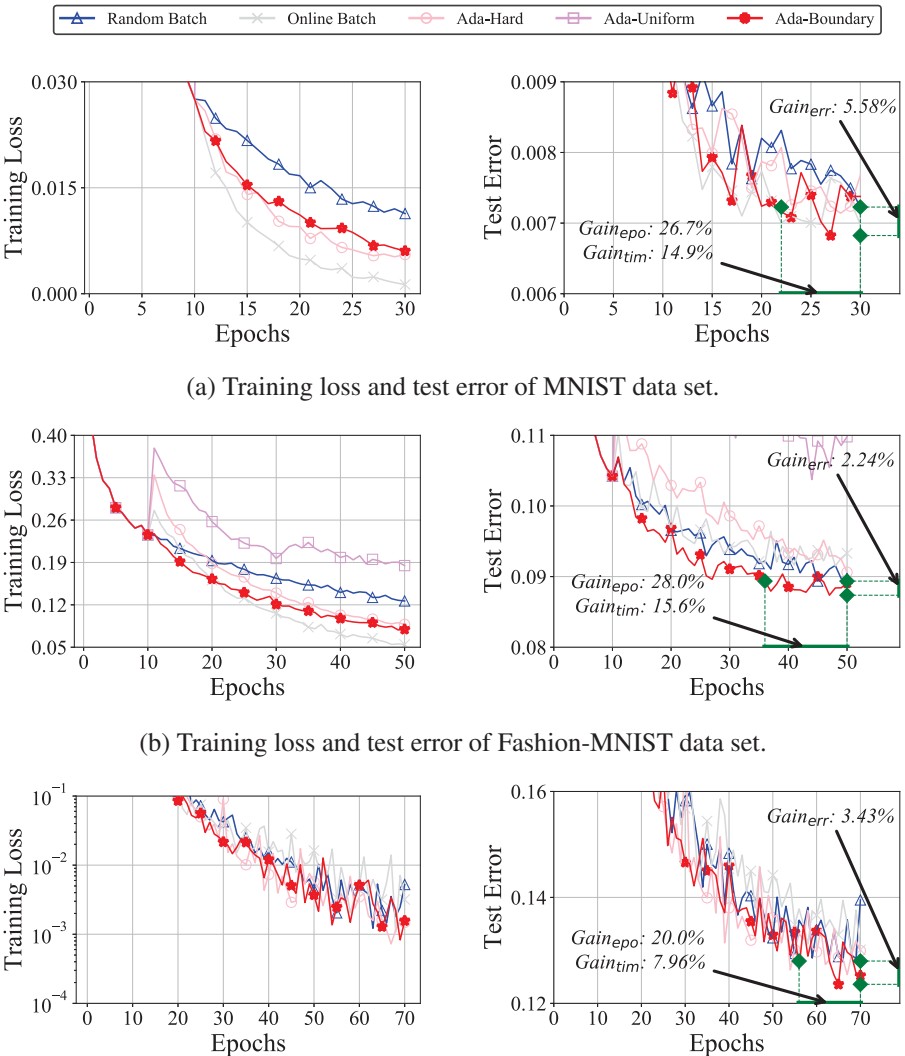

(a) Training loss and test error of MNIST data set.

(b) Training loss and test error of Fashion-MNIST data set.

(c) Training loss and test error of CIFAR-10 data set.

Figure 10: Convergence curves using the **momentum** optimizer for Figure 6.

