# OpenReview forum: "Ada-Boundary: Accelerating the DNN Training via Adaptive Boundary Batch Selection"
_ICLR.cc/2019/Conference_

### Official Review · AnonReviewer2 · 2018-10-31
**Decent paper, but little added insight beyond "Active Bias"**

**Rating:** 5
**Confidence:** 4

**Review:**

This paper attempts to speed up convergence of deep neural networks by intelligently selecting batches. The experiments show this method works moderately well.

This paper appears quite similar to the recent work "Active Bias" [1].
The motivation for the technique and setting appear very similar, while the details of the techniques are different. Unfortunately, this is not mentioned in the related work, or even cited.

When introducing a new method, it is important that design choices are principled, have theoretical guidance, or are experimentally verified against similar design choices. Without one of these, the methods become arbitrary and it is unclear what causes better performance. Unfortunately, this paper makes several choices, about an uncertainty function, the probability distribution, the discretization, and the algorithm (when to update) that appear rather arbitrary. For instance, the uncertainty function is a signed standard deviation of the softmax output. While there are a variety of uncertainty functions, such as entropy and margin, a new seemingly arbitrary uncertainty function is introduced.

The experiments are good but could be designed a bit better. For instance, it is unclear if the gains are because of lower asymptotic error or because of faster convergence. The learning curves are stopped too early, while the test error is still dropping quickly.

In summary, it is not clear if this paper adds any insight beyond "Active Bias".

[1] Active Bias: Training More Accurate Neural Networks by Emphasizing High Variance Samples. 2017. Haw-Shiuan Chang, Erik Learned-Miller, Andrew McCallum.

---

> ### Author Response · Authors · 2018-11-06
> **Response**
>
> We would like to thank AnonReviewer2 for the helpful comments. We will revise our paper as soon as possible to reflect your comments. Below are our responses.
>
> 1. Related to “Active Bias” paper:
> - Thanks again for mentioning the recent related work that we did not consider. We have just read “Active Bias” paper and agree that their motivation is similar to ours: “Active Bias” also claims that we should prefer uncertain samples, not easy or hard samples. However, we would like you to know that the goal of our paper differs greatly from that of “Active Bias”. “Active Bias” gives more weight to the samples with large variance of prediction probabilities for its true label during training, and finally succeeds to train more accurate and robust model. On the other hand, we select the samples near the decision boundary and focus on training the model quickly in a given time. Therefore, insights of our paper for fast training are clearly different from those of “Active Bias” (“Active Bias” paper did not compare the convergence speed at all). We will discuss this in the related work section.
>
> 2. Learning curves are stopped too early:
> - As mentioned earlier, the main contribution of this paper is accelerating convergence speed (or saving training time), rather than reducing the final test error. In this situation, since the training step is fully converged at the end of training, we thought that it would be more appropriate to show the clear difference by stopping the training procedure early.
>
> 3. Asymptotic error vs faster convergence:
> - We tried to avoid the gains from lower asymptotic error. We are sorry for the missing detail in our paper. We thought that random weights for initialization cause such asymptotic gains, so we updated all strategies for the same random batch with the same initial weights during \gamma epochs called warm-up period. Therefore, as shown in Figure 6(b), the convergence curve for training/test is the same in all methods. We believe that such weight sharing technique can mitigate the asymptotic gains in training.

---

> > ### Comment · AnonReviewer2 · 2018-12-03
> > **Comparison to Active Bias**
> >
> > It is known that emphasizing uncertain examples or examples close to the decision boundary can improve asymptotic error (not only Active Bias paper, but a few other papers as well). I don't see how the warm-starting removes this effect. Thus, without showing the full learning curves, it is unclear if the better error for a fixed number of iterations is because of faster convergence or because of lower asymptotic error. If it's the former, that could be interesting. However, I'm unconvinced that it's not the latter which would mean the contribution is the same as Active Bias.

---

> ### Author Response · Authors · 2018-11-24
> **Revised version uploaded.**
>
> Our revised version has been uploaded. We have made a clear statement of the similarity and difference with "Active Biase", and have clarified the reason for early stopping. Also, we have clearly described the missing experimental design (e.g., why not set \gamma = 1?, why share initial parameters?), which is devised to show that our performance gains comes from the fast convergence. More details can be found in the brief summary of changes above.
>
> Thank you again for the comments that helped us to revise the paper.

---

### Official Review · AnonReviewer3 · 2018-11-03
**An adaptive batch normalization approach with limited technical novelty.**

**Rating:** 5
**Confidence:** 3

**Review:**

The paper introduces an adaptive importance sampling strategy, as opposed to uniform sampling, for batch normalization. The key idea is to assign higher importance to those correctly classified training samples with relatively smaller soft-max prediction variance, hopefully to push the deep nets to learn faster from uncertain samples near the decision boundary. Experimental results on several benchmark datasets (MNIST, CIFAR-10) and commonly used deep nets (LeNet, ResNet) are reported to show the power of boundary batch selection in improving the overall training efficiency.

The paper is clearly presented and the numerical results are mostly easy to access. My main concern is about the novelty of technical contribution which is mainly composed by two: 1) a prediction variance based importance sampling strategy for batch selection and 2) an empirical study the show the merits of approach. Concerning the first contribution, the idea of defining boundary samples according to prediction variance looks fairly common, if not superficial, in modern machine learning. The way of defining the sampling probability (see Eq. 4 & 5) follows largely the rank-based method (Loshchilov and Hutter 2016) with slight modifications. The numerical study shows some promise of the proposal on several relatively easy data sets. However, as a practical paper, the numerical results could be much more supportive if more challenging data sets (e.g., ImageNet) are included for evaluation.

Pros:

-The method is well motivated and clearly presented.
- The paper is easy to follow.


Cons:

-  The overall contribution is incremental with limited novelty.
- As a practical paper, the numerical study falls short in evaluation on large-scale data.

---

> ### Author Response · Authors · 2018-11-08
> **Response**
>
> We would like to thank AnonReviewer3 for the useful comments. We would like to respond to your main concern.
>
> 1. Novelty of this paper:
> - As you mentioned, the idea of defining boundary samples using “prediction variance” looks fairly common. However, we hope you look into our contribution as the insight that boundary samples should be preferred in training step for acceleration, rather than the simplicity of the methodology. As for the methodology, we tried various and complex methods (e.g., a modified pseudo loss of adaboost, adaptive weight scaling for each sample, ...) to decide the boundary samples, but we selected “prediction variance” since the performance was generally good. Also, we thought that the simplicity of the method is the best advantage for practical use.
> - For similar reasons, we selected the quantization-based method to compute the selection probability of each sample.
>
> 2. Large-scale data:
> - As a practical paper, we agree that it would be better if we showed the performance of the proposed method for a more challenging data set (e.g., ImageNet). However, in our limited experimental environment, handling such data set like ImageNet required too much effort, so we focused on showing the performance details for the three benchmark data sets (MNIST, Fashion-MNIST, and CIFAR-10). In our future work, we are planning to add the result performed on more diverse data sets and neural architectures.

---

> ### Author Response · Authors · 2018-11-24
> **Revised version uploaded.**
>
> Our revised version has been uploaded. As a practical paper, we have performed additional experiments on two challenging data sets (CIFAR-100 and Tiny-ImageNet) to supplement the lack of experiments in our paper. More details can be found in the brief summary of changes above.
>
> We hope you look into our contribution as the insight that boundary samples should be preferred in training step for acceleration, rather than the simplicity of the methodology.
>
> Thank you again for the comments that helped us to revise the paper.

---

### Official Review · AnonReviewer1 · 2018-11-03
**The proposed construction for more effective sampling during DNN training is conceptually nice and has the potential for wide impact, but the paper does not provide clear evidence that value is gained.**

**Rating:** 5
**Confidence:** 4

**Review:**

The paper describes a sampling distribution construction over examples from which to draw mini-batches to train multi-classification models. A distance function on examples is described wherein an example's current (softmax) label probabilities and correctness are taken into account. The bounded distance function supports quantization of example distances and then subsequent sampling from an exponentially decaying probability mass function defined over the binned examples. Results from experiments implementing the proposed method and some baselines on three image classification datasets are provided.

Clearly, any generic improvement to training DNN's has the potential for far-reaching impact. I thought the exposition was fairly clear and appreciated how the introductory sections provided an intuitive understanding of e.g., the differences between the proposed method and the method of Loshchilov and Hutter (2016). The relative conceptual simplicity of the proposed method is a clear positive. The experimental methodology and results are my biggest issues with the paper. The experimental evaluation suggests the proposed method was run 3 times, one for each value of the selection pressure parameter. Then, the best run was selected for comparison. This suggests the proposal is not practical. For results, the benefit of the proposed method is only clearly apparent in one of the three experiments (Fashion MNIST). In the MNIST case, the proposed method does not seem to improve upon the online batch method. For CIFAR-10, where a good case for the proposed method could have been made since the architecture is more complex and potentially more difficult to train, the improvement seems slight. Moreover, it isn't clear whether a relevant baseline was included (see second question below). Also, at least some discussion of computational cost incurred by the method should have been provided. Even better would be to include results wrt/ wall clock training time.

Questions/Comments:

Why not set \gamma = 1? Having a larger value seems to run counter to making training faster. Technically, to use the proposed method, all of the examples need to be processed only once before the distance-based sampling distribution can be utilized.

Does the random method of the paper denote uniformly at random from the entire dataset per batch or sequential batches from a pre-shuffled dataset per epoch? The distinction is important as Loshchilov and Hutter (2016) report that the latter method performs better than their online batch method on CIFAR-10.

ADA-easy seems to be an irrelevant baseline given the context of the paper.

The phrase "learner's level" is used multiple times, but not defined.

The average is reported in the convergence curves, but shouldn't the variance be reported as well?

Perhaps the selection pressure parameter can be annealed as performed in Loshchilov and Hutter (2016)?

---

> ### Author Response · Authors · 2018-11-06
> **Response**
>
> We would like to thank AnonReviewer1 for the valuable questions and comments. We will revise our paper as soon as possible to reflect your comments. Below are our responses.
>
> 1. The selection pressure parameter:
>  -  To be precise, the best selection pressure value is “100”, so all experiments in our paper used this value. We conducted an additional experiment to find which value of s_e={10, 100, 1000} is the best for acceleration, and in general “100” showed the best performance in our three benchmark data sets. We will make this clear in our revised version.
>
> 2. Improvement on CIFAR-10:
>  - In the CIFAR-10 data set (Figure 6(c)), the convergence improvement may seem insignificant, but numerically the proposed method reduced 24.3% of the total number of epochs (i.e., 24.3% reduction of the number of parameter updates). Also, even considering the additional computation cost of the proposed method, we saved 14.0% of training time. We think that such improvement is meaningful. Please note that it was obtained without any modification of learning rate, batch size and loss function, which have a great effect on training speed.
>
> 3. Computational cost of proposed method:
>  - We are sorry for the missing of the computational cost incurred by our method. We thought that it can be indirectly inferred by subtracting Gain_epo and Gain_tim. For example, in the CIFAR-10 data set (Figure 6(c)), the 10.3% gap between Gain_epo and Gain_tim was incurred by the extra cost of our method.
>
> 4. The large value of \gamma:
>  - Technically, your comment is right. But, we tried to exclude the impact of random initial weights to highlight the effect of batch selection strategies. As you can see in Figure 6(b), all methods shared the model weights until reaching \gamma epochs. We thought the weight sharing during large \gamma epochs can remove the impact of random initial weights on training step.
>
> 5. The random method:
>  - Random method denotes uniformly at random from entire data set per batch. But, in our experimental setting for CIFAR-10, online batch method was worse than random method. We have not yet compared the convergence speed with shuffle method.
>
> 6. Variance:
>  - As we mentioned, we tried to reduce the variance of each run by sharing the weights during \gamma epochs, so there is no significant difference for each run. Also, we concerned that figure 6-9 would be too messy.
>
> 7. Annealed:
>  - Of course, the selection pressure can be annealed.

---

> ### Author Response · Authors · 2018-11-24
> **Revised version uploaded.**
>
> Our revised version has been uploaded. We are grateful for your deep understanding of our paper. Your questions and comments have been a great help in improving our paper, so we have tried to reflect your comments as many as possible. You can see more details of the changes that reflecting your comments above.
>
> Thank you again for the comments that helped us to revise the paper.

---

### Author Response · Authors · 2018-11-24
**To all reviewers: Our paper is updated! (Brief summaries are listed below.)**

Dear reviewers. We would like to thank for all valuable comments.
We have made a revised version that reflects your comments as many as possible.

First of all, we clarify the novelty of our paper.
- An in-depth analysis of why existing hard batch selection does not always accelerate training.
- The insight that boundary samples should be preferred in training step for acceleration.
- The simplicity of our method "Ada-Boundary" for practical use.

Here is a brief summary of the changes we made reflecting your comments:

R1. The best selection pressure parameter s_e (by anonreviewer1)
- In line 11-12 of Section 4.1, we have explicitly mentioned that we set the selection pressure to be 100, which is the best value found from s_e in {10, 100, 1000}.

R2. The computational cost of the proposed method (by anonreviewer1)
- In Appendix D, we have discussed the additional computational cost incurred by our method. Also, we have included the wall-clock time of all experiments.

R3. The large value of \gamma (by anonreviewer1)
- In line 13-15 of Section 4.1, we have explained the reason: "Technically, a small \gamma was enough to warm-up, but to reduce the performance variance caused by randomly initialized parameters, we used the larger \gamma and shared model parameters for all strategies during the warm-up period."

R4. The random method (by anonreviewer1)
- In line 2 of Section 4.2, we have clearly denoted that the random batch selection selects the next batch uniformly at random from the entire data set.

R5. learner's level (by anonreviewer1)
- As you mentioned, the term "the learner's level" is very confusing to understand, so we have written the term more clearly in our paper: "the learner's level" -> "the learning progress of the model"

R6. More challenging datasets (by anonreviewer3)
- As a practical paper, we have performed additional experiments on more challenging data sets: CIFAR-100 (100 classes) and Tiny-ImageNet (200 classes). The details have been discussed in Appendix C.

R7. Related to "Active Bias" (by anonreviewer2)
- At the end of the third paragraph of Section 5, we have made a clear statement of the similarity and difference with "Active Bias" paper. The main contributions of our paper lie on training faster, but that of "Active Bias" lies on training more accurate and robust model.

R8. Learning curves are stopped too early (by anonreviewer2)
- In line 10-11 of Section 4.1, we have explained why we early stopped: Our main contribution is fast training, so "early stopping" helps to clearly show the difference in convergence speed.

Besides, we have discussed the impact of the selection pressure s_e on training in Appendix C.1.
We will do our best by the end of the rebuttal period. Thank you.

---

### Meta-Review · Area_Chair1 · 2018-12-11
**Could be improved with better explanation of the insights and experiment design.**

**Confidence:** 4
**Recommendation:** Reject

**Metareview:**

This paper introduced an adaptive importance sampling strategy to select mini-batches to speed up the convergence of network training. The method is well motivated and easy to follow.

The main concerns raised by the reviewers are limited novelty of the proposed simple idea compared to related recent work, and moderate empirical performance.

The authors argue that the particular choice of the adaptive sampling method comes after trying various methods. I believe providing more detailed discussion and comparison with different methods together with the "active bias" paper would help the readers appreciate the insights conveyed in this paper.

The authors provide some additional experiments in the revision. It would make the whole experiment section a lot stronger and convincing if the authors could run more thorough experiments on extra challenging datasets and include all the results int the main text.

Additional experiment to clarify the merit of the proposed method on either faster convergence or lower asymptotic error would also improve the contribution of this paper.